# Cross-Document Event Coreference Resolution on Discourse Structure

**Xinyu Chen, Sheng Xu, Peifeng Li** and **Qiaoming Zhu**
School of Computer Science and Technology
Soochow University, China
{xychennlper, sxu}@stu.suda.edu.cn, {pfli, qmzhu}@suda.edu.cn

## Abstract

Cross-document event coreference resolution (CD-ECR) is a task of clustering event mentions across multiple documents that refer to the same real-world events. Previous studies usually model the CD-ECR task as a pairwise similarity comparison problem by using different event mention features, and consider the highly similar event mention pairs in the same cluster as coreferent. In general, most of them only consider the local context of event mentions and ignore their implicit global information, thus failing to capture the interactions of long-distance event mentions. To address the above issue, we regard discourse structure as global information to further improve CD-ECR. First, we use a discourse rhetorical structure constructor to construct tree structures to represent documents. Then, we obtain shortest dependency paths from the tree structures to represent interactions between event mention pairs. Finally, we feed the above information to a multi-layer perceptron to capture the similarities of event mention pairs for resolving coreferent events. Experimental results on the ECB+ dataset show that our proposed model outperforms several baselines and achieves the competitive performance with the start-of-the-art baselines.

## 1 Introduction

In real-world texts, there are usually a large number of sentences that describe the same event in reality, and an event will be mentioned repeatedly in multiple documents. When multiple event mentions (an instance of a specific event in texts) point to the same event ontology, these event mentions are coreferent. Event coreference resolution is useful for many natural language processing (NLP) applications, such as information extraction (Liu et al., 2017), topic detection (Wayne, 1998), and question answering (Weissenborn et al., 2017). Depending on whether the event mentions are in the same document, this task can be further divided into within-document (WD-ECR) and cross-document (CD-ECR) event coreference resolution. This paper focuses on the cross-document task CD-ECR.

Events mainly consist of triggers and arguments. Since triggers are the main words that can most clearly express the occurrence of events, each event can be represented as its corresponding trigger. Consider the following two event mentions as examples:

*S1: The court would hand down a ruling on whether the former president will remain **detained** for three more months before the current extension expires.*

*S2: The former president **detainment** was previously extended for three months in July.*

The event triggers in the event sentences *S1* and *S2* are "detained" and "detainment", respectively. Although these two triggers have different forms, they both refer to the same judicial type $Arrest$. Therefore, "detained" in *S1* and "detainment" in *S2* have a coreference relationship and can be aggregated to form a coreferent chain.

Previous studies on WD-ECR or CD-ECR typically took a pair of event mentions with their event sentences as input and then predicted whether they are coreferent using a binary classifier (Chen and Ji, 2009; Lu et al., 2016; Lu and Ng, 2017). To this end, an important step in their models was to extract discriminative features of event mentions and then used encoding method to represent event mentions as vectors. Simultaneously, almost all previous studies considered the task of event coreference resolution as a similarity model and focused on how to calculate the feature similarities between event mention pairs (Liu et al., 2014; Lu and Ng, 2017). Besides, some studies also focused on data augmentation (Nguyen et al., 2016; Choubey and Huang, 2018; Huang et al., 2019; Barhom et al., 2019; Fang and Li, 2020), which boost event coreference resolution on additional raw data.

In the CD-ECR task, there is a limitation that

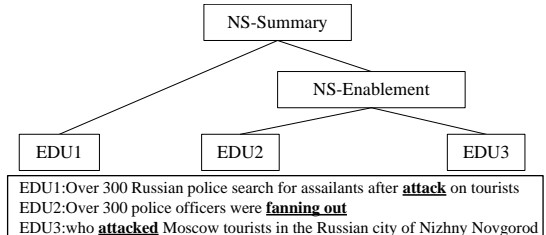

Figure 1: An example of discourse tree. The three underlined words are event mention triggers.

two event mentions in a pair may be scattered across one or more documents and have a long distance. Previous studies on both CD-ECR and WD-ECR have attempted to extract the context features, which only represented event mentions from the local perspective (Kenyon-Dean et al., 2018; Barhom et al., 2019) and the information from the global perspective was neglected.

To address the above issue, we introduce discourse structure to extract global information of event mentions and then apply them to CD-ECR. Discourse structure is a tree structure (called "discourse tree" below). Generally, discourse tree is constructed by discourse rhetorical structure (Mann and Thompson, 1998) (DRS) constructor, which aims to represent input text as a tree structure like the example in Figure 1, where the leaf nodes are elementary discourse units (EDU) and it has smaller fine-grained units than event sentences. The discourse tree can clearly express the rhetorical relation and nuclearity relation (e.g., NS-Summary) of each EDU, which can provide useful information for associating those long-distance event mention pairs.

On the other hand, the DRS constructor generally performs tree construction on single document. In the CD-ECR task, if two event mentions in the same document, we can directly input the document to DRS constructor to obtain discourse tree, and then extract the global information of the two event mentions. However, if the query is a cross-document event mention pair, and the models send their documents to the DRS constructor separately, the global information of this event mention pair cannot be obtained directly. Therefore, we also propose a construction strategy of cross-document discourse tree, which is helpful to obtain the global information of cross-document event mention pairs. We summarize the contributions of our work as follows.

- We extract global information from discourse

tree which can provide useful information for associating those long-distance event mention pairs.

- We propose a strategy for constructing the cross-document discourse tree. To the best of our knowledge, we are the first to apply the cross-document discourse tree to the CD-ECR task.

- Our model outperforms several baselines and achieves the competitive performance with the start-of-the-art baseline.

## 2 Related Work

Research on event coreference resolution mainly draws on the method of entity coreference resolution, which aims to resolve noun phrases/mentions for entities (Raghunathan et al., 2010; Ng, 2010; Durrett and Klein, 2013; Lee et al., 2017; Joshi et al., 2019). Since event mentions have more complex structures than entity mentions, event coreference resolution is a more challenging task than entity coreference resolution (Yang et al., 2015).

Early research on event coreference resolution mainly applied machine learning methods (Chen and Ji, 2009; Bejan and Harabagiu, 2010; Liu et al., 2014). Some researchers incorporated various kinds of regular methods into machine learning methods and improved the performance of event coreference resolution (Nicolae and Nicolae, 2006; Sangeetha and Arock, 2012; Liu et al., 2018). Since these methods relied heavily on manual annotation features, some studies paid more attention to raw text event coreference resolution (Araki and Mitamura, 2015; Peng et al., 2016; Lu et al., 2016; Chen and Ng, 2016; Lu and Ng, 2017).

Recently, deep learning work has been applied to both WD-ECR (Nguyen et al., 2016; Choubey and Huang, 2018; Fang et al., 2018; Huang et al., 2019; Cheng et al., 2019; Lu et al., 2020; Choubey et al., 2020; Lu and Ng, 2021) and CD-ECR (Kenyon-Dean et al., 2018; Barhom et al., 2019; Zeng et al., 2020; Cattan et al., 2021; Yu et al., 2022). In WD-ECR, Krause et al. (2016) used convolutional neural networks to mine event features for event coreference resolution. Huang et al. (2019) incorporated argument compatibility knowledge from a large number of the unlabeled corpus. Lu and Ng (2021) investigated span-based models for event coreference resolution. In CD-ECR, argument information was introduced into event representations

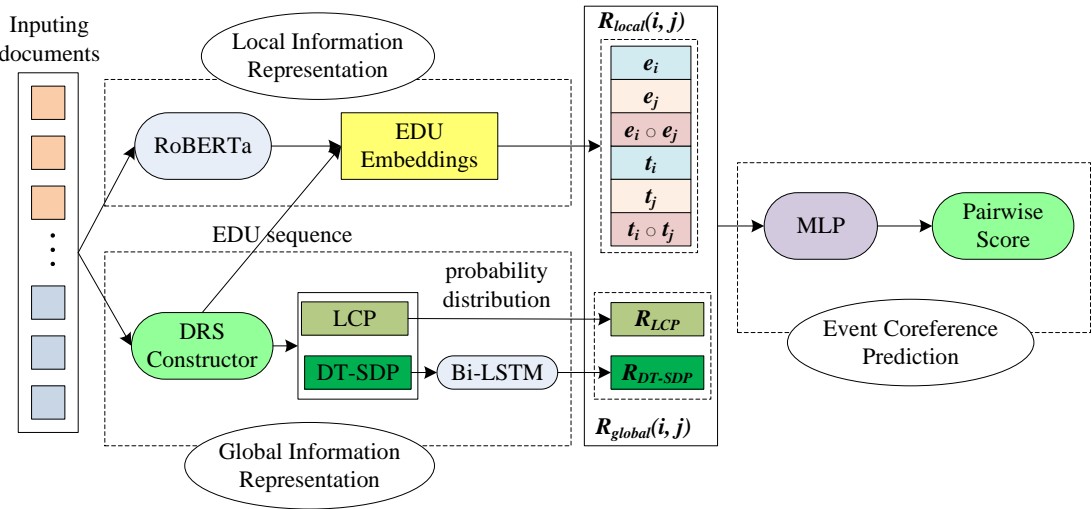

Figure 2: The overall framework of our CD-ECR model.

(Barhom et al., 2019; Zeng et al., 2020; Yu et al., 2022), Barhom et al. (2019) jointly learned entity and event coreference resolution and leveraged predicate-argument structures. Zeng et al. (2020) integrated event-specific paraphrases and argument-aware semantic embeddings for CD-ECR. Yu et al. (2022) augmented pairwise representation with structured argument features to improve CD-ECR performance. Caciularu et al. (2021) pretrained a cross-document language model via sets of related documents for CD-ECR. Held et al. (2021) extracted event mentions features from the local perspective and trained a fine-grained classifier to improve CD-ECR performance.

Compared with previous deep learning work for CD-ECR, we present a novel global information representation for event mentions to enhance the interaction between long-distance event mentions.

## 3 Model

Formally, given a set of documents $D = \{d_1, d_2, ..., d_{|D|}\}$, whose element $d_i$ consists of a series of sentences $\{s_{i1}, s_{i2}, ..., s_{i|d_i|}\}$, and a set of event mentions $M = \{m_1, m_2, ..., m_{|M|}\}$, these event mentions in $M$ can be distributed in different documents in $D$. CD-ECR aims to discover the event mentions in $M$ from different documents in $D$ that refer to the same event ontology in the real world and gather them into the same cluster. ECB+ is the most popular corpus for CD-ECR and our work is also performed on this corpus.

Figure 2 shows an overview of our CD-ECR model, which includes three main components: 1) Local Information Representation (LIR) to obtain

local perspective information of event mentions from EDU embeddings, 2) Global Information Representation (GIR) to extract global information representation from the discourse tree constructed by the DRS Constructor, 3) Event Coreference Prediction (ECP) to receive the global and local representations of event mention pairs and predict the probability that two event mentions are coreferent.

### 3.1 Local Information Representation

Generally, previous pairwise model on the CD-ECR task took the concatenated vector $R(m_i, m_j) = [v(m_i), v(m_j), v(m_i) \circ v(m_j)]$ as the base representation of event pairs, where $\circ$ is element-wise multiplication, the $v(m_i)$ and $v(m_j)$ are feature vector of the event mention $m_i$ and $m_j$, respectively. They obtained the feature vector $v(\cdot)$ through encoding the event sentence by a pre-trained language model (e.g., BERT (Xu et al., 2019), RoBERTa (Liu et al., 2019)) and then consider the word embedding of trigger or trigger context tokens as the feature vector $v(\cdot)$.

Different from previous work, we incorporate EDU representation in $R(m_i, m_j)$ where event mention located. Specifically, we first encode each document using RoBERTa$_{LARGE}$ inspired by Cattan et al. (2021), which splits long documents into non-overlapping segments of up to 512 word-piece tokens and encodes them independently. Differently, when the document exceeds 512 tokens, in order to preserve the integrity of the EDU information where the trigger is located, we just split it into multiple segments of entire EDUs and not just 512 tokens.

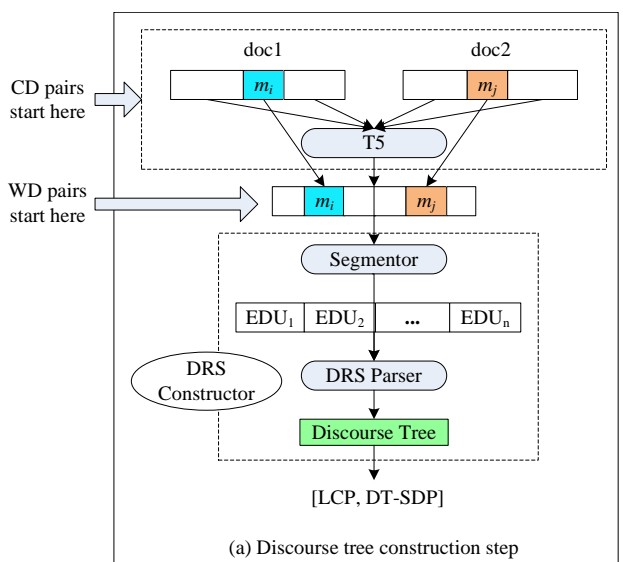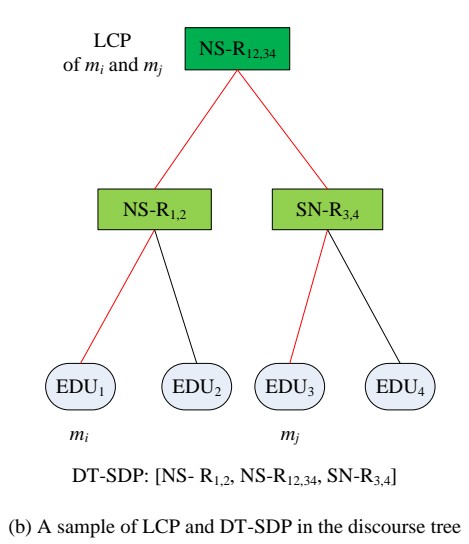

(a) Discourse tree construction step

(b) A sample of LCP and DT-SDP in the discourse tree

Figure 3: Architecture of DRS constructor and a sample of global information representation.

After encoded by RoBERTa, we extract the trigger pair feature $t(m_i, m_j)$ and the EDU pair feature $e(m_i, m_j)$ from word embeddings for the event mention pair $(m_i, m_j)$, the local information representation $R_{local}(i, j)$ is obtained as follows.

$$
\begin{aligned}
t(m_i, m_j) &= [t_i, t_j, t_i \circ t_j], \\
e(m_i, m_j) &= [e_i, e_j, e_i \circ e_j], \quad (1) \\
R_{local}(i, j) &= [t(m_i, m_j), e(m_i, m_j)],
\end{aligned}
$$

where $t_i$, $t_j$ denotes the trigger tokens embeddings of $m_i$, $m_j$, and $e_i$, $e_j$ denotes the EDU tokens embeddings of $m_i$, $m_j$.

### 3.2 Global Information Representation

GIR aims to obtain the global information representation of event mention pair from discourse tree. Figure 3(a) shows the discourse tree construction step for both within-document (WD) and cross-document (CD) event mention pairs.

Generally, a DRS constructor consists of two main components: Segmentor and DRS parser. The segmentor receives an article with several sentences and splits it into a set of EDU sequences [EDU$_1$, EDU$_2$,...,EDU$_n$], where all EDUs are leaf nodes of the discourse tree. Then, the DRS parser predicts a specific rhetoric and nuclearity relation between two adjacent EDUs, and then forms a superior DU, which links to others EDUs or DUs to obtain a discourse tree.

Specifically, we first prepare the input data for the DRS constructor. Since the two event mentions in a given input event mention pair $(m_i, m_j)$ may come from different documents, we process it in two ways as shown in Figure 3(a). If $(m_i, m_j)$ is a within-document event mention pair, we feed whole document directly to the DRS constructor to obtain a discourse tree. If $(m_i, m_j)$ is a cross-document event mention pair, we extract the two event sentences that $m_i$ and $m_j$ are located respectively, then the other texts are sent to T5$_{BASE}$ (Raffel et al., 2020) for compression. The compressed texts are concatenated with two event sentences by their origin order to form the input document for the DRS constructor. Finally, the cross-document discourse tree is obtained. In our work, we utilize the state-of-the-art discourse rhetoric structure parser (DRS) (Zhang et al., 2021) to construct discourse trees for the CD-ECR task-specific corpus.

Shortest Dependency Path (SDP) is widely used in various NLP tasks (Xu et al., 2015; Cheng and Miyao, 2017) to capture crucial interaction information between sentences, we combine it with discourse tree for our CD-ECR task, called DT-SDP. As shown in Figure 3(b), we assume that the event mentions $m_i$ and $m_j$ are located in EDU$_1$ and EDU$_3$, respectively, and the shortest path between them is marked in red. We express nodes on the red line as the sequence [NS-R$_{1,2}$, NS-R$_{12,34}$, NS-R$_{3,4}$] to represent DT-SDP. All non-leaf nodes contain the probability distributions information of the rhetoric and nuclearity relation obtained by the DRS parser. Since different event mention pairs can obtain different length of DT-SDP, we apply Bi-LSTM network to encode variable length sequence DT-SDP, and take the output of last hidden layer as DT-SDP representation, denoted as $R_{DT-SDP}$.

Additionally, we also extract the lowest common parent node LCP of two EDUs that the event mentions $m_i$ and $m_j$ are located from the discourse tree (as shown by dark green node in the Figure 3(b)). The probability distributions information of LCP is denoted as $R_{LCP}$. We finally obtain the global information representation $R_{global}(i, j)$ by concatenating $R_{LCP}$ and $R_{DT-SDP}$ as follows.

$$R_{global}(i, j) = [R_{LCP}, R_{DT-SDP}]. \quad (2)$$

### 3.3 Event Coreference Prediction

In this stage, we have obtained the global and local information representations $R_{global}(i, j)$ and $R_{local}(i, j)$ of the event mention pair $(m_i, m_j)$. We fuse the two features and send them to the multi-layer perceptron (MLP) and sigmoid activation function to obtain the coreference score $S$ as follows.

$$\theta = MLP(R_{global}(i, j), R_{local}(i, j)), \quad (3)$$

$$S = Sigmoid(\theta). \quad (4)$$

### 3.4 Training and Inference

During training, we apply dropout in Bi-LSTM and MLP networks, the training objective is to minimize the binary cross-entropy loss $L$ as follows.

$$L = -\frac{1}{N}\sum_{i=1}^{N}[y_i log\hat{y}_i + (1-y_i)log(1-\hat{y}_i)], \quad (5)$$

where $N$ is the size of event mention pair samples and $y \in \{0, 1\}$ is a pair label.

During inference, we first reproduce the topic predictor of Barhom et al. (2019) to perform document clustering, and take event mention pairs in the same document cluster as candidate coreferent pairs. We then send these pairs to our CD-ECR model to obtain the coreference score, and consider the pairs whose score $> 0.5$ as coreferent pairs, otherwise as non-coreferent. To handle the pairwise event coreference predictions, we perform best-first clustering (Huang et al., 2019) on the pairwise scores to build the coreferent event clusters.

## 4 Experimentation

### 4.1 Experimental Settings

**Dataset** Following previous work (Cybulska and Vossen, 2014), we use the ECB+ dataset to train and test our model, which is the largest and most popular dataset for CD-ECR. ECB+ is extended from ECB (Bejan and Harabagiu, 2010), which

|  | Train | Dev | Test |
|---|---|---|---|
| Topics | 25 | 8 | 10 |
| Documents | 574 | 196 | 206 |
| Sentences | 1037 | 346 | 457 |
| Event mentions | 3808 | 1245 | 1780 |
| Event Singletons | 1116 | 280 | 623 |
| Event Clusters | 1527 | 409 | 805 |

Table 1: ECB+ statistics. We follow the data split by Cybulska and Vossen (2015): train: 1, 3, 4, 6-11, 13-17,19-20, 22, 24-33; dev: 2, 5, 12, 18, 21, 23, 34, 35; test:36-45. Event Clusters include singletons.

annotated different but similar events as subtopics for each ECB topic. We use gold event mentions for both training and evaluation. The detailed statistics are shown in Table 1.

**Metrics** Following the previous work (Barhom et al., 2019; Cattan et al., 2021; Yu et al., 2022), we use MUC (Vilain et al., 1995), $B^3$ (Bagga, 1998), and CEAF$_e$ (Luo, 2005) to evaluate the performance of our model and also report the CoNLL scores, which is the average of the above three metrics. Among them, MUC is based on event links to evaluate the performance of the model, $B^3$ compensates for MUC's neglected evaluation of non-coreferent events by using event nodes as the computational target. CEAF$_e$ is similar to $B^3$, adding entities to evaluate the performance of event coreference resolution. The comprehensive use of the above three metrics and CoNLL can more objectively measure model performance.

**Hyper Parameters** We use the pre-trained language model RoBERTa$_{LARGE}$ to embed event mentions with 1024 dimensions, the training epoch of our model is set to 10, the learning rate is set to $10^{-5}$, Adam optimizer is used to update the parameters. The minimal and maximum output length of T5$_{BASE}$ generator are set to 20% and 50% of the input text, respectively. Additionally, in order to make a fair comparison with the baseline of Caciularu et al. (2021), which uses stronger encoder Longformer$_{BASE}$, we also use the Longformer$_{BASE}$ to embed event mentions with 768 dimensions for resolving coreferent events.

### 4.2 Experimental Results

To verify the effectiveness of our model, we conduct the following strong baselines.

1) Barhom et al. (2019), which jointly learns entity and event coreference resolution and leverages predicate-argument structures;

| Model | MUC | | | B$^3$ | | | CEAF$_e$ | | | CoNLL |
|---|---|---|---|---|---|---|---|---|---|---|
| | P | R | F1 | P | R | F1 | P | R | F1 | F1 |
| Barhom et al. (2019) | 84.5 | 77.6 | 80.9 | 85.1 | 76.1 | 80.3 | 73.8 | 81.0 | 77.3 | 79.5 |
| Zeng et al. (2020) | 85.6 | 89.3 | 87.5 | 77.6 | 89.7 | 83.2 | 84.5 | 80.1 | 82.3 | 84.3 |
| Cattan et al. (2021) | 81.9 | 85.1 | 83.5 | 82.7 | 82.1 | 82.4 | 78.9 | 75.2 | 77.0 | 81.0 |
| Caciularu et al. (2021)* | 89.2 | 87.1 | 88.1 | 87.9 | 84.9 | 86.4 | 81.2 | 83.3 | 82.2 | 85.6 |
| Held et al. (2021) | 88.1 | 87.0 | 87.5 | 87.7 | 85.6 | 86.6 | 85.8 | 80.3 | 82.9 | 85.7 |
| Yu et al. (2022) | 85.1 | 88.1 | 86.6 | 84.7 | 86.1 | 85.4 | 79.6 | 83.1 | 81.3 | 84.4 |
| Ours(RoBERTa) | 85.9 | 88.6 | 87.2 | 85.4 | 87.8 | 86.6 | 83.7 | 82.8 | 83.2 | 85.7 |
| Ours(Longformer)* | 87.2 | 89.4 | **88.3** | 86.4 | 88.3 | **87.3** | 84.0 | 83.2 | **83.6** | **86.4** |

Table 2: Performance comparision of different models on the ECB+ dataset, where "*" indicates that the models use LongFormer as their encoders and the other models use BERT/RoBERTa as their encoders.

2) Zeng et al. (2020), which integrates event-specific paraphrases and argument-aware semantic embeddings for CD-ECR;

3) Cattan et al. (2021), which develops an end-to-end baseline for CD-ECR;

4) Caciularu et al. (2021), which pretrains a language model via a sets of related documents for CD-ECR. Longformer was used for their encoder.

5) Held et al. (2021), which extracts event mentions features from the local perspective and trained a fine-grained classifier for CD-ECR.

6) Yu et al. (2022), which augments pairwise representation with structured argument features to improve CD-ECR performance.

Table 2 reports the performance of the above six baselines and our model on ECB+ with encoder RoBERTa and Longformer, and the results show that our model (Longformer) significantly (P<0.01) outperforms the best Held et al. (2021), with the improvement of 0.7 in the average score CoNLL, and our model (RoBERTa) achieve the competitive result with them. This result indicates the effectiveness of our proposed model in resolving coreferent event.

Formally, the common part of Barhom et al. (2019), Zeng et al. (2020) and Yu et al. (2022) is that their input feature of event mention pairs can be represented as $R(m_i, m_j) = [v(m_i), v(m_j), v(m_i) \circ v(m_j), f(m_i, m_j)]$, where $f(m_i, m_j)$ is additional pairwise feature. Barhom et al. (2019) trains entity and event coreference together and takes argument features as $f(m_i, m_j)$, which outperforms several early CD-ECR models (Cybulska and Vossen, 2015; Kenyon-Dean et al., 2018). Zeng et al. (2020) not only uses the argument features but also integrates event-specific paraphrases, significantly improving CoNLL by

4.8 over Barhom et al. (2019). Yu et al. (2022) augments pairwise representation with structured argument features, improving CoNLL by 4.9 over Barhom et al. (2019) and achieving competitive performance with Zeng et al. (2020). This suggests that argument features are crucial for resolving coreferent events. However, using argument features to calculate event similarity is from local perspective, ignoring the global features of events, which leads to the poor performance on long-distance event mentions. Our model uses discourse trees to capture interactions between event mentions, thus enhancing the representation of event information and improving CD-ECR performance.

Compared with Cattan et al. (2021), who develop an end-to-end baseline for CD-ECR and only use RoBERTa to encode event mentions without using other features, our model improves the CoNLL score by 4.7. It also shows the effectiveness of global information in the discourse tree, since our baseline is also RoBERTa.

Compared with Caciularu et al. (2021), who employed Longformer as the encoder. Since Longformer is capable of encoding entire documents, it outperforms other encoders in the event coreference resolution task. For fair comparison, we replace our text encoder RoBERTa with Longformer and the experimental results shows that our model outperforms Caciularu et al. (2021) with average CoNLL score improvements of 0.8.

Compared with Held et al. (2021), both our model (RoBERTa) and Held et al. (2021) achieve the same CoNLL score 85.7. Held et al. (2021) focuses on extracting event mentions features from the local perspective and trains a fine-grained classifier. Compared with them, our model pays more

| Model | | | MUC | | | $B^3$ | | | $CEAF_e$ | | | CoNLL |
|---|---|---|---|---|---|---|---|---|---|---|---|---|
| | | | P | R | F1 | P | R | F1 | P | R | F1 | F1 |
| Ours | Corpus | singletons+ | 85.9 | 88.6 | 87.2 | 85.4 | 87.8 | 86.6 | 83.7 | 82.8 | 83.2 | **85.7** |
| | | singletons- | 85.9 | 88.6 | 87.2 | 74.5 | 76.1 | 75.3 | 57.4 | 76.9 | 65.7 | **76.4** |
| | Topic | singletons+ | 82.0 | 84.6 | 83.3 | 75.4 | 71.8 | 73.6 | 82.1 | 81.8 | 81.9 | **79.6** |
| | | singletons- | 82.0 | 84.6 | 83.3 | 57.8 | 58.0 | 57.9 | 48.7 | 65.4 | 55.8 | **65.7** |
| Cattan | Corpus | singletons+ | 81.9 | 85.1 | 83.5 | 82.7 | 82.1 | 82.4 | 78.9 | 75.2 | 77.0 | 81.0 |
| | | singletons- | 81.9 | 85.1 | 83.5 | 70.2 | 70.8 | 70.5 | 52.3 | 68.2 | 59.2 | 71.1 |
| | Topic | singletons+ | 76.3 | 80.1 | 78.1 | 71.7 | 77.4 | 74.5 | 77.8 | 73.1 | 75.4 | 76.0 |
| | | singletons- | 76.3 | 80.1 | 78.1 | 54.1 | 63.4 | 58.4 | 44.2 | 56.3 | 49.5 | 62.0 |

Table 3: Performance comparison of corpus/topic level and with(singletons+)/without(singletons-) singletons.

attention to extracting event mention features from the global perspective, which uses discourse tree to capture the features of long-distance event mention pairs. In a word, the performance improvement of our model and Held et al. (2021) can be owned to the intensive study of global and local features, respectively.

Comapred with our model Ours(RoBERTa) using RoBERTa as encoder, Ours(Longformer) using Longformer improves CoNLL by 0.7. This indocates that LongFomer is a stronger encoder than RoBERTa in the CD-ECR task.

In the above baselines, Cattan et al. (2021) evaluate their model not only at the corpus level with singletons, but also at the topic level without singletons. We also report the result on these experiment settings. The performance comparison with Cattan et al. (2021) at the corpus/topic level with (singletons+)/without (singletons-) singletons are shown in Table 3. The results show that our model significantly outperforms Cattan et al. (2021) on both corpus level and topic level with/without singletons on all metrics.

## 5 Analysis

### 5.1 Impact of Global Information

We conduct ablation experiments to further evaluate the contribution of global information from discourse tree and design the following simplified models.

1) Baseline(RoBERTa), which resolves coreferent events using local feature $R_{local}(m_i, m_j)$ only;

2) +WD-LCP, which adds the global feature $R_{global}(m_i, m_j)$ of within-document event mention pairs to baseline 1). However, $R_{global}(m_i, m_j)$ only contains $R_{LCP}$;

3) +CD-LCP, which adds the global feature $R_{global}(m_i, m_j)$ of cross-document event mention pairs to baseline 2). However, $R_{global}(m_i, m_j)$ only contains $R_{LCP}$;

4) +SDP, which adds the feature $R_{DT-SDP}$ to baseline 3).

The results are shown in Table 4. Since the process of constructing cross-document discourse trees is more complicated than that of within-document discourse tree (mainly in the data preparation stage), we compare the performance of adding the cross-document LCP information separately due to the other influencing factors such as data noise.

Compared with baseline(RoBERTa), +WD-LCP improves CoNLL by 1.2 on the ECB+ dataset using the within-document LCP. This preliminarily shows the effectiveness of the LCP information in discourse tree. Compare with the baseline +WD-LCP and +CD-LCP, the introduction of the cross-document LCP leads to an improvement of 1.6 on CoNLL. This indicates that LCP can alleviate the long-distance problem which is poorly handled by local information representation. The comparison on +CD-LCP and +DT-SDP shows the significant improvement of 1.8 on CoNLL and indicates that DT-SDP is crucial to capture the interaction between long-distance event mentions.

### 5.2 Analysis on Construction Strategy for Cross Document Discourse Structure

In order to deeply analyse the impact of different construction strategies of the cross-document DRS tree, we design two other cross-document DRS construction strategies for comparison with the strategy in our model as follows.

1) ES-comp, which takes event sentence and other sentences compressed by T5 model as constructor input;

2) ES-only, which takes only event sentence as constructor input without any other text;

| Model | MUC | | | B³ | | | CEAFₑ | | | CoNLL |
|---|---|---|---|---|---|---|---|---|---|---|
| | P | R | F1 | P | R | F1 | P | R | F1 | F1 |
| Baseline(RoBERTa) | 82 | 85.2 | 83.6 | 82.9 | 82.2 | 82.5 | 75.5 | 79.2 | 77.3 | 81.1 |
| +WD-LCP | 83.1 | 85.2 | 84.1 | 83.2 | 82.4 | 82.8 | 79.2 | 80.6 | 79.9 | 82.3 |
| +CD-LCP | 84.8 | 87.8 | 86.3 | 84.9 | 83.3 | 84.1 | 82 | 80.7 | 81.3 | 83.9 |
| +DT-SDP | 85.9 | 88.6 | **87.2** | 85.4 | 87.8 | **86.6** | 83.7 | 82.8 | **83.2** | **85.7** |

Table 4: An ablation study for discourse structure.

| Strategy | | MUC | | | B³ | | | CEAFₑ | | | CoNLL |
|---|---|---|---|---|---|---|---|---|---|---|---|
| | | P | R | F1 | P | R | F1 | P | R | F1 | F1 |
| ES-comp | w/ DT-SDP | 85.9 | 88.6 | **87.2** | 85.4 | 87.8 | **86.6** | 83.7 | 82.8 | **83.2** | **85.7** |
| | w/o DT-SDP | 84.8 | 87.8 | 86.3 | 84.9 | 83.3 | 84.1 | 82.0 | 80.7 | 81.3 | 83.9 |
| ES-only | w/ DT-SDP | 84.8 | 86.6 | 85.7 | 84.0 | 82.7 | 83.3 | 78.3 | 81.3 | 79.8 | 82.9 |
| | w/o DT-SDP | 84.5 | 86.3 | 85.4 | 83.9 | 82.5 | 83.2 | 78.0 | 81.1 | 79.5 | 82.7 |
| Full-doc | w/ DT-SDP | 85.2 | 86.7 | 85.9 | 84.2 | 86.1 | 85.1 | 81.1 | 82.2 | 81.6 | 84.2 |
| | w/o DT-SDP | 83.7 | 85.6 | 84.6 | 80.5 | 84.2 | 82.3 | 77.5 | 80.2 | 78.8 | 81.9 |

Table 5: Results using different construction strategies of cross-document DRS tree.

3) Full-doc, which concatenates two documents that contain the event mention pairs and then sends them to the constructor.

The results are shown in Table 5. Regardless of whether DT-SDP information is introduced, the construction strategy ES-comp outperforms the two other strategies. If we exclude DT-SDP, the Full-doc strategy achieves the worst performance due to the data noise introduced by irrelevant texts. The ES-only strategy achieves the second worst performance, which is slightly better than Full-doc. However, it is fails to outperform ES-comp because it has no other contextual information to guide it.

Comparing each strategy with (w/) or without (w/o) DT-SDP, we can see that the ES-only strategy has the least performance improvement when DT-SDP is included because the input text contains only two event sentences, which is too short, and the depth and breadth of the constructed discourse tree are too small, leading to the short length of the shortest path between two EDUs. Using the node sequence to represent the path may only include common parent nodes, which tends to exclude DT-SDP. In contrast, the performance of the Full-doc strategy achieves the most improvement. This is because the long input text increases the depth and breadth of the discourse tree, which also increases the length of the shortest path and enriches the representation of DT-SDP. However, the existence of a large amount of noise also limits the increase in performance so that it does not outperform ES-

comp. It shows that ES-comp is a compromise strategy, because taking event sentences and other compressed irrelevant texts as constructor input can increase the positive impact so as to achieve the best performance.

### 5.3 Case Study

In this subsection, we give the examples to analyse the effectiveness of global information and the strategy ES-comp in our model. Considering the following two documents *A* and *B* as examples, where the bolded texts represent event sentences (*S1* and *S4*, respectively), and the underlined words indicate event mentions, the goal of our model is to predict whether the event mentions "winning" and "chosen" are coreferent.

*A: (S1)* **Smith, 26, who played a young political researcher in the show, will become the biggest star of all after winning the role of the 11th Doctor.** *(S2) Speaking to The Guardian, Buchan said his old co-star would make an excellent Doctor Who. "It s a sublime bit of casting. He's got that huge hair, a twinkle in his eye - Matt's the king of geek chic. He is possibly going to be one of the best Doctors we've ever had."*

*B: (S3) 26-year-old Matt Smith has been cast as the next incarnation of the Doctor. Users on the Facebook Doctor Who forum that I frequent mostly had the same reaction: " Who 's Matt Smith?" (S4)* **The guy is relatively unknown and the skeptics wondered if the right person was chosen.** *(S5) After all, everyone speculated that Paterson Joseph,*

*who had appeared on a couple episodes of the show, was going to be the next Doctor and here we get some no-name.*

We utilize the ES-comp approach to generate a discourse tree. Initially, we compress the three non-bolded sections *S2*, *S3*, and *S5*, resulting in the compressed texts *C1*, *C2*, and *C3*, respectively, as follows.

*C1: It's a sublime bit of casting.*

*C2: Matt Smith has been cast as the next incarnation of the Doctor. "Who 's Matt Smith?"*

*C3: we get Paterson Joseph.*

Then, the compressed documents *A* and *B* are concatenated as [*S1, C1, C2, S4, C3*] and then are fed to the DRS Constructor. The resulting EDU sequence is displayed as follows.

*$EDU_1$: Smith, 26,*

*$EDU_2$: who played a young political researcher in the show,*

*$EDU_3$: will become the biggest star of all*

*$EDU_4$: after winning the role of the 11th Doctor*

*$EDU_5$: It's a sublime bit of casting.*

*$EDU_6$: "Matt Smith has been cast as the next incarnation of the Doctor.*

*$EDU_7$: " Who's Matt Smith ? "*

*$EDU_8$: The guy is relatively unknown*

*$EDU_9$: and the skeptics wondered*

*$EDU_{10}$: if the right person was chosen*

*$EDU_{11}$: we get Paterson Joseph .*

Figure 4 illustrates the simplified discourse tree produced by the ES-comp strategy. Notably, since we solely examine the coreferent relation between "winning" and "chosen", Figure 4 excludes irrelevant nodes like $EDU_{11}$. Furthermore, $EDU_1$, $EDU_2$, and $EDU_3$ are merged into $EDU_{1-3}$ since these three nodes are not part of DT-SDP. The DT-SDP in this case is [NS-Elaborate, NS-Elaborate, NS-Cause, SN-Span, SN-Cause].

Directly measuring the similarity between $EDU_4$ and $EDU_{10}$ to predict coreference relation would result in misidentification due to their differing contexts. Since the rhetorical relation nodes

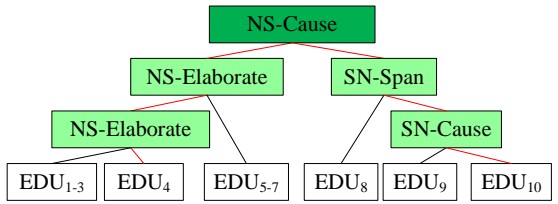

Figure 4: The cross-document discourse tree constructed by the strategy ES-comp.

"NS-Elaborate" in DT-SDP associate $EDU_4$ with $EDU_{1-3}$, $EDU_{1-3}$ can introduce the agent "Smith" of the event mention "winning" to $EDU_4$. "Smith" can also be passed to the event mention "chosen" in $EDU_{10}$ using DT-SDP. This makes "Smith" apparent in the tokens of "winning" and "chosen" and facilitates accurate prediction.

In this case, if we use the strategy ES-only, the node "NS-Elaborate" will be excluded in the obtained DT-SDT and the common argument "Smith" cannot be perceived. Finally, if we use the Full-doc strategy, the pair still gets low coreference score, because the tree structure becomes complicated and the redundant information interferes with the model's perception of argument information.

## 6 Conclusion

In this paper, we propose a cross-document event coreference resolution model. The novelty of our model is twofold. First, we introduce the discourse structure that represents global information for pairs of event mentions to provide support for event coreference resolution. Second, we propose a strategy for constructing cross-document discourse trees, which allows cross-document coreferent event mentions to be easily identified by the model. Experimental results on the ECB+ dataset show that our proposed model outperforms several baselines. In the future, we plan to extend our task to cross-modal event coreference resolution.

### Limitations

Our method still suffers from several shortcomings. First, we focus only on the event coreference resolution step using annotated event mentions. The upstream task event detection is also critical for event coreference resolution. Second, although the introduction of the discourse tree and global information has a great performance boost, the inference time of the DRS constructor is long. Last but not least, there is still room for optimization in the strategy of cross-document DRS construction.

### Acknowledgements

The authors would like to thank the three anonymous reviewers for their comments on this paper. This research was supported by the National Natural Science Foundation of China (Nos. 62276177, 61836007, and 62376181), and Project Funded by the Priority Academic Program Development of Jiangsu Higher Education Institutions (PAPD).

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
