# OpenReview forum: "Cross-Document Event Coreference Resolution on Discourse Structure"
_EMNLP/2023/Conference — EMNLP 2023 Main_

### Official Review · Reviewer_48CY · 2023-08-03

**Soundness:** 4

**Excitement:**

3: Ambivalent: It has merits (e.g., it reports state-of-the-art results, the idea is nice), but there are key weaknesses (e.g., it describes incremental work), and it can significantly benefit from another round of revision. However, I won't object to accepting it if my co-reviewers champion it.

**Missing References:**

- CDLM: Cross-Document Language Modeling (Caciularu et al., 2021)
- Focus on what matters: Applying Discourse Coherence Theory to Cross Document Coreference (Held et al., 2021).
- Contrastive Representation Learning for Cross-Document Coreference Resolution of Events and Entities (Hsu and Horwood, 2022)
- What happens before and after: Multi-Event Commonsense in Event Coreference Resolution (Ravi et al., 2023)

**Paper Topic And Main Contributions:**

This paper presents a new model for cross-document event coreference resolution that enriches the pairwise scorer with discourse information. Specifically, the scorer takes as input local information (embeddings of the event trigger and embeddings of the EDU), and global discourse information (lowest parent node of the two EDUs and the shortest path between the two EDUs). To find global discourse information between two event mentions in different documents, the authors first combine the documents by taking the full sentences of the event mentions and compressing the other parts of the texts using a T5 model, then run a discourse parser on this new “document”.

The model achieves a new state-of-the-art on ECB+, a popular dataset for the task. In addition, the authors provide an ablation study that shows the benefit of the different components of the pairwise scorer.

**Questions For The Authors:**

The sentence in lines 225-231 lacks clarity, the EDU sequence is concatenated to what? Do you mean that you concatenated the EDU and encode all EDUs using RoBERTa in addition to encoding each document with RoBERTa? Also, I don’t really understand the constraints in page 3 equation 1. How the sum of all EDUs is less than 512 but the sum of the length of all EDUs except the last is greater than 512?

**Reasons To Accept:**

- The paper proposes a nice approach to integrate discourse signals into a cross-document event coreference model, providing substantial gain over the baseline (+4.6 F1).


- The ablations were thoroughly conducted, showing the effect of (almost) each component, both in the pairwise scorer and the strategy for running a discourse parser on two documents.

**Reasons To Reject:**

- The paper doesn’t compare the proposed model to actual state-of-the-art models (that outperform the proposed model by ~1 F1 point):
CDLM: Cross-Document Language Modeling (Caciularu et al., 2021)
Focus on what matters: Applying Discourse Coherence Theory to Cross Document Coreference (Held et al., 2021).
While it may not be necessary for the model to surpass the current SOTA, the paper should still discuss the advantages of the proposed model over the SOTA, (e.g., robustness, runtime, etc). Unfortunately, the paper fails to cite these relevant papers.


- The model is evaluated only at the corpus level with singletons. It would be nice to also evaluate the model at the topic level without singletons, as proposed in “Realistic Evaluation Principles for Cross-document Coreference Resolution (Cattan et al., 2021)”.  A few recent works have adopted this evaluation methodology, so it’ll be great to also compare your results to these models.



- The model is evaluated only on a single dataset.

**Reproducibility:**

3: Could reproduce the results with some difficulty. The settings of parameters are underspecified or subjectively determined; the training/evaluation data are not widely available.

**Reviewer Confidence:**

4: Quite sure. I tried to check the important points carefully. It's unlikely, though conceivable, that I missed something that should affect my ratings.

**Typos Grammar Style And Presentation Improvements:**

- Line 22: perpceptron → perceptron
- Line 228: missing space before the open parentheses
- Line 284: missing space before the open parentheses

---

> ### Author Rebuttal · Authors · 2023-08-28
>
> Thank you very much for your valuable comments.
>
> **Q1: The paper doesn’t compare the proposed model to actual state-of-the-art models (that outperform the proposed model by ~1 F1 point): CDLM: Cross-Document Language Modeling (Caciularu et al., 2021) Focus on what matters: Applying Discourse Coherence Theory to Cross Document Coreference (Held et al., 2021).**
>
> **A1**: We will add Caciularu et al. (Findings of EMNLP 2021) and Held et al. (EMNLP 2021) to the "Related Work" section and add both papers to our baselines in the revised paper.
>
> Caciularu et al. (2021) was not used as baselines in the original paper because they employed stronger pre-trained model Longformer as the encoder, whereas our model and the baselines used RoBERTa as the encoder. However, utilizing LongFormer requires significant GPU memory (approximately 44GB), necessitating a powerful graphics card such as the A100-80G for efficient operation. This requirement presents a challenge to researchers and institutions with limited computing resources.
>
> It would be unfair to directly compare our RoBERTa-based model to those using Longformer. For fair comparison with Caciularu et al. (2021), we replace our text encoder RoBERTa with Longformer and the experimental results are showed in the following table. The table shows that our model outperforms Caciularu with average CoNLL score improvements of 0.8.
>
> We also add Held et al. (2021) as a baseline, which used fine-tuning RoBERTa. Both our model and Held et al. (2021) achieve the same CoNLL score 85.7. Held et al. (2021) focused on extracting event mentions features from the local perspective and trained a fine-grained classifier. Compared with them, our model pays more attention to extracting event mention features from the global perspective, which uses DT-SDP and LCP to capture the features of long-distance event mention pairs. In a word, the performance improvement of our model and Held et al. (2021) can be owned to the intensive study of global and local features, respectively.
>
> We will include these results in our revised paper.
>
> | **Model**        |  | **$MUC$** |  | | **$B^3$** |  |  | **$CEAF_e$** | | **$CoNLL$** |
> |:----------------:|:----:|:-------:|:----:|:----:|:------:|:----:|:----:|:---------:|:----:|:---------:|
> |                  | P    | R       | F1   | P    | R      | F1   | P    | R         | F1   | F1        |
> | Caciularu(CDLM)  | 89.2 | 87.1    | 88.1 | 87.9 | 84.9   | 86.4 | 81.2 | 83.3      | 82.2 | 85.6      |
> | Held             | 88.1 | 87.0    | 87.5 | 87.7 | 85.6   | 86.6 | 85.8 | 80.3      | 82.9 | 85.7      |
> |Ours(RoBERTa-based)|85.9|88.6|87.2|85.4|87.8|86.6|83.7|82.8|83.2|85.7|
> | Ours(Longformer-based)             | 87.2 | 89.4    | **88.3** | 86.4 | 88.3   | **87.3** | 83.2 | 84.0      | **83.6** | **86.4**      |
>
>
> **Q2: The model is evaluated only at the corpus level with singletons.**
>
> **A2**: The performance comparison with Cattan et al at the topic level with (singletons+) and without (singletons-) singletons are shown in following table. The results in this table show that our model outperforms Cattan et al. (2021) on both corpus level and topic level with/without singletons.
>
> |    | **Model** |         |  | **$MUC$** |  |  | **$B^3$** |  |  | **$CEAF_e$** |  | **$CoNLL$** |
> |:------:|:---------:|:-----------:|:----:|:-------:|:--------:|:--------:|:------:|:--------:|:--------:|:---------:|:--------:|:----------:|
> |        |           |             | P    | R       | F1       | P        | R      | F1       | P        | R         | F1       | F1         |
> | Ours   | Corpus    | singletons+ | 85.9 | 88.6    | 87.2     | 85.4     | 87.8   | 86.6     | 83.7     | 82.8      | 83.2     | 85.7       |
> |        |           | singletons- | 85.9 | 88.6    | 87.2     | 74.5     | 76.1   | 75.3     | 57.4     | 76.9      | 65.7     | 76.4       |
> |        | Topic     | singletons+ | 82.0 | 84.6    | 83.3     | 75.4     | 71.8   | 73.6     | 82.1     | 81.8      | 81.9     | 79.6       |
> |        |           | singletons- | 82.0 | 84.6    | 83.3     | 57.8     | 58.0   | 57.9     | 48.7     | 65.4      | 55.8     | 65.7       |
> | Cattan | Corpus    | singletons+ | 81.9 | 85.1    | 83.5     | 82.7     | 82.1   | 82.4     | 78.9     | 75.2      | 77.0     | 81.0       |
> |        |           | singletons- | 81.9 | 85.1    | 83.5     | 70.2     | 70.8   | 70.5     | 52.3     | 68.2      | 59.2     | 71.1       |
> |        | Topic     | singletons+ | 76.3 | 80.1    | 78.1     | 71.7     | 77.4   | 74.5     | 77.8     | 73.1      | 75.4     | 76.0       |
> |        |           | singletons- | 76.3 | 80.1    | 78.1     | 54.1     | 63.4   | 58.4     | 44.2     | 56.3      | 49.5     | 62.0       |
>
> **Q3: The model is evaluated only on a single dataset.**
>
> **A3**: Yes, it is better to evaluate our model on multiple datasets. However, ECB+ is the most popular dataset for cross-document event coreference resolution, most previous work only evaluated their performance on ECB+. Following previous work for comparison, we also evaluate our model on ECB+.
>
> **Q4: The sentence in lines 225-231 lacks clarity, the EDU sequence is concatenated to what? Do you mean that you concatenated the EDU and encode all EDUs using RoBERTa in addition to encoding each document with RoBERTa? I don’t really understand the constraints in page 3 equation 1. How the sum of all EDUs is less than 512 but the sum of the length of all EDUs except the last is greater than 512?**
>
> **A4**: These two constraints are used to truncate a document with the limitation of 512 tokens.
>
> We will rewrite this parts as follows:
>
> However, to maintain the EDU information's integrity where the trigger is located, we combined the EDU sequence obtained from the segmentor in the DRS Constructor (outlined in Section 3.2) to create a DU that does not exceed a length of 512.
>
> (1)	If N (the token number in a document) is less than or equal to 512, we encode the full document by RoBERTa directly.
>
> (2)	If N is greater than 512, we first identify the EDU containing t512 (assuming it is the (q+1)-th EDU), and then encode the [EDU1, EDU2,...,EDUq] using RoBERTa.
>
> (3) We update the value of N to the number of tokens of the remaining documents and renumber them.
>
> (4) We repeat (1)-(3) until all tokens in a document are encoded.

---

### Official Review · Reviewer_T5Xb · 2023-08-04

**Soundness:** 4

**Excitement:**

4: Strong: This paper deepens the understanding of some phenomenon or lowers the barriers to an existing research direction.

**Missing References:**

Focus on what matters: Applying Discourse Coherence Theory to Cross Document Coreference
https://aclanthology.org/2021.emnlp-main.106.pdf

Contrastive Representation Learning for Cross-Document Coreference Resolution of Events and Entities
https://aclanthology.org/2022.naacl-main.267.pdf

CDLM: Cross-Document Language Modeling
https://aclanthology.org/2021.findings-emnlp.225.pdf

**Paper Topic And Main Contributions:**

The paper focuses on the problem "Cross-Document Event Coreference Resolution". The paper addresses the lack of global information embedded in the final representation for the prediction. So the paper proposed a discourse parser to construct a discourse tree to represent each document. Then its shortest path between two mentions, and the lowest common parent node are extracted from the discourse tree to encode global information.  The paper shows the performance improvement for some metrics, The paper also provided the ablation study and analysis of the construction strategy.

**Questions For The Authors:**

Q1. L208 In the local information, it is very common to have $v(m_i) - v(m_j)$ as another representation. Is there any reason why the paper did not use it?

**Reasons To Accept:**

1. The paper proposed to use global information to provide more information to the prediction which is new for this event coreference problem.
2. The paper is well-written and easy to follow.
3. The paper provided an analysis of the parsing strategy to obtain the tree for the global information and an ablation study.

**Reasons To Reject:**

1. The paper missed some stronger baselines that have similar/better performance on some metrics. This may justify the SOTA achievement of this paper. E.g. Held et at 2021 (https://aclanthology.org/2021.emnlp-main.106.pdf) got $B^3=86.6$ which is equal to the $B^3$ performance of this paper. Caciularu et al. 2021 (https://aclanthology.org/2021.findings-emnlp.225.pdf) have $MUC=88.1 $ which is greater than 87.2 of this paper and CONLL=85.6 which is equivalent to 85.7 of this paper.
2. The method in this paper depends heavily on the discourse parser, the paper has not reported the effect of the performance of the discourse parser accuracy on the final performance.

**Reproducibility:**

3: Could reproduce the results with some difficulty. The settings of parameters are underspecified or subjectively determined; the training/evaluation data are not widely available.

**Reviewer Confidence:**

4: Quite sure. I tried to check the important points carefully. It's unlikely, though conceivable, that I missed something that should affect my ratings.

**Typos Grammar Style And Presentation Improvements:**

Equation 2: The equations should be aligned using the "=". using the latex environment "aligned" with "&" before the "=" will make them aligned.
L214. "e.g." should have been "e.g.," (with the comma)
L284: a space between the word "tasks" and the parenthesis is needed.
Table 1,2,3,4: The model names should be aligned on the left.
Figure 2b: The example is not very good as it does not show the lowest common parent node clearly.
Figure 4: The bottom borders of the EDU's boxes are missing.

---

> ### Author Rebuttal · Authors · 2023-08-28
>
> Thank you very much for your valuable comments.
>
> **Q1: The paper missed some stronger baselines that have similar/better performance on some metrics.**
>
> **A1**: We will add Caciularu et al. (Findings of EMNLP 2021) and Held et al. (EMNLP 2021) to the "Related Work" section and add both papers to our baselines in the revised paper.
>
> Caciularu et al. (2021) was not used as baselines in the original paper because they employed stronger pre-trained model Longformer as the encoder, whereas our model and the baselines used RoBERTa as the encoder. However, utilizing LongFormer requires significant GPU memory (approximately 44GB), necessitating a powerful graphics card such as the A100-80G for efficient operation. This requirement presents a challenge to researchers and institutions with limited computing resources.
>
> It would be unfair to directly compare our RoBERTa-based model to those using Longformer. For fair comparison with Caciularu et al. (2021), we replace our text encoder RoBERTa with Longformer and the experimental results are showed in the following table. The table shows that our model outperforms Caciularu with average CoNLL score improvements of 0.8.
>
> We also add Held et al. (2021) as a baseline, which used fine-tuning RoBERTa. Both our model and Held et al. (2021) achieve the same CoNLL score 85.7. Held et al. (2021) focused on extracting event mentions features from the local perspective and trained a fine-grained classifier. Compared with them, our model pays more attention to extracting event mention features from the global perspective, which uses DT-SDP and LCP to capture the features of long-distance event mention pairs. In a word, the performance improvement of our model and Held et al. (2021) can be owned to the intensive study of global and local features, respectively.
>
> We will include these results in our revised paper.
>
> | **Model**        |  | **$MUC$** |  | | **$B^3$** |  |  | **$CEAF_e$** | | **$CoNLL$** |
> |:----------------:|:----:|:-------:|:----:|:----:|:------:|:----:|:----:|:---------:|:----:|:---------:|
> |                  | P    | R       | F1   | P    | R      | F1   | P    | R         | F1   | F1        |
> | Caciularu(CDLM)  | 89.2 | 87.1    | 88.1 | 87.9 | 84.9   | 86.4 | 81.2 | 83.3      | 82.2 | 85.6      |
> | Held             | 88.1 | 87.0    | 87.5 | 87.7 | 85.6   | 86.6 | 85.8 | 80.3      | 82.9 | 85.7      |
> |Ours(RoBERTa-based)|85.9|88.6|87.2|85.4|87.8|86.6|83.7|82.8|83.2|85.7|
> | Ours(Longformer-based)             | 87.2 | 89.4    | **88.3** | 86.4 | 88.3   | **87.3** | 83.2 | 84.0      | **83.6** | **86.4**      |
>
>
> **Q2: The method in this paper depends heavily on the discourse parser, the paper has not reported the effect of the performance of the discourse parser accuracy on the final performance.**
>
> **A2**: Since there is no discourse relation annotated in the ECB+ dataset, we did not report the performance of discourse parsing in our paper. At present, the performance of discourse parsing remains low. In our paper, we utilize Zhang's system to create discourse trees, achieving F1-scores of 71.8, 59.5, and 47.0 for Span, Nuclearity, and Relation tasks on the RST dataset.
>
>
> **Q3: L208 In the local information, it is very common to have $v_{m_i}-v_{m_j}$ as another representation. Is there any reason why the paper did not use it?**
>
> **A3**: $v_{m_i}-v_{m_j}$ typically represents the Euclidean distance between two event mentions, while $v_{m_i} \circ v_{m_j}$ typically represents cosine similarity. Cosine similarity places greater emphasis on the difference in direction between the vectors of event mentions. Maintaining consistency in the direction of event mention vectors is important in determining whether two event mentions are coreferent, which Euclidean distance fails to capture.

---

### Official Review · Reviewer_fuqf · 2023-08-04

**Soundness:** 4

**Excitement:**

4: Strong: This paper deepens the understanding of some phenomenon or lowers the barriers to an existing research direction.

**Missing References:**

Avi Caciularu, Arman Cohan, Iz Beltagy, Matthew Peters, Arie Cattan, and Ido Dagan. 2021. CDLM: Cross-Document Language Modeling. In Findings of the Association for Computational Linguistics: EMNLP 2021, pages 2648–2662, Punta Cana, Dominican Republic. Association for Computational Linguistics.

**Paper Topic And Main Contributions:**

This paper studies the task of cross-document event coreference. They propose methods to incorporate the discourse structure of documents for both within-document and cross-document mention pairs. They use a combination of two discourse related signals, lowest common parent (LCP) and shortest dependency path in the discourse tree (DT-SDP). For the cross-document mention pairs, they compress the two documents except for the two focus sentences using a T5-based system and create a meta document.

The experiments on the standard ECB+ dataset show the effectiveness of the proposed method, especially the DT-SDP module. Interestingly, the T5-based compression of cross-document context is quite effective and outperforms the use of full document texts.

**Questions For The Authors:**

1. Results in Table 4 are interesting. Compressing the rest of the document seems to be quite effective. I wonder what information this compressed document includes, and why it's better than full documents. The author (s) attribute this to noise in line 515, but some additional analysis here can be useful.
2. In the ECB+ dataset, only few sentences from each document were annotated with coreference relations. To construct the discourse structure, did the author(s) use the full documents or only these annotated sentences?
3. Following up on the above point, how would the discourse structure based method deal with longer documents. The related work CDLM (cited above) works on longer context inputs, so how do they compare against each other in long document settings.

**Reasons To Accept:**

1. The paper studies an important research question about integrating global discourse information into the pairwise coreference resolution task. This is well-motivated for the task of cross-document coreference resolution. The methods presented are simple and innovative and provide strong results over the listed baselines.
2. The ablations studies are quite thorough. The paper showcases the effectiveness of each module, within-doc LCP, cross-doc LCP, and the DT-SDP. The use of T5-based compression for cross-document context is simple and effective.
3. The paper is mostly well-written and provides enough details about the methods described in the work.

**Reasons To Reject:**

1. The paper completely misses out a comparing against important prior work. Cross-document document modeling has previously been studied for this task (Caciularu et al., Findings 2021 (https://aclanthology.org/2021.findings-emnlp.225/)) and the paper doesn't provide any mention of this work. The results presented in this work are comparable to those in Caciularu et al., Findings 2021.
2. The qualitative analysis presented in section 5.3 is hard to follow. The intuition behind the use of discourse structure can be described better in the paper.

**Reproducibility:**

4: Could mostly reproduce the results, but there may be some variation because of sample variance or minor variations in their interpretation of the protocol or method.

**Reviewer Confidence:**

4: Quite sure. I tried to check the important points carefully. It's unlikely, though conceivable, that I missed something that should affect my ratings.

---

> ### Author Rebuttal · Authors · 2023-08-28
>
> Thank you very much for your valuable comments.
>
> **Q1: The paper completely misses out a comparing against important prior work. Cross-document document modeling has previously been studied for this task (Caciularu et al., Findings 2021 (https://aclanthology.org/2021.findings-emnlp.225/)) and the paper doesn't provide any mention of this work.**
>
> **A1**: We will add Caciularu et al. (Findings of EMNLP 2021) to the "Related Work" section and add it to our baselines in the revised paper.
>
> This work was not used as baselines in the original paper because it employed Longformer as the encoder, whereas our model and baselines used RoBERTa as the encoder. Since Longformer is capable of encoding entire documents, it outperforms other encoders in the event coreference resolution task. However, utilizing the document-level event encoder LongFormer requires significant GPU memory (approximately 44GB), necessitating a powerful graphics card such as the A100-80G for efficient operation. This requirement presents a challenge to researchers and institutions with limited computing resources.
>
> It would be unfair to directly compare our RoBERTa-based model to those using the stronger pre-training language model Longformer. For fair comparison, we replace our text encoder RoBERTa with Longformer and the experimental results are showed in the following table. The table shows that our model outperforms Caciularu with average CoNLL score improvements of 0.8. We will include these results in our revised paper.
>
> | **Model**        |  | **$MUC$** |  |  | **$B^3$** | |  | **$CEAF_e$** |  | **$CoNLL$** |
> |:----------------:|:----:|:-------:|:----:|:----:|:------:|:----:|:----:|:---------:|:----:|:---------:|
> |                  | P    | R       | F1   | P    | R      | F1   | P    | R         | F1   | F1        |
> | Caciularu(CDLM)  | 89.2 | 87.1    | 88.1 | 87.9 | 84.9   | 86.4 | 81.2 | 83.3      | 82.2 | 85.6      |
> | Ours             | 87.2 | 89.4    | **88.3** | 86.4 | 88.3   | **87.3** | 83.2 | 84.0      | **83.6** | **86.4**      |
>
> **Q2: The qualitative analysis presented in section 5.3 is hard to follow. The intuition behind the use of discourse structure can be described better in the paper.**
>
> **A2**: We will revise this section as follows:
>
> Considering the following two documents A and B as examples, where the bolded texts represent event sentences (S1 and S4, respectively), and the underlined words indicate event mentions,  the goal of our model is to predict whether the event mentions “winning” and “chosen” are coreferent.
>
> A: (S1)**Smith, 26, who played a young political researcher in the show, will become the biggest star of all after winning the role of the 11th Doctor.** (S2) Speaking to The Guardian, Buchan said his old co-star would make an excellent Doctor Who. ``It s a sublime bit of casting. He's got that huge hair, a twinkle in his eye - Matt's the king of geek chic. He is possibly going to be one of the best Doctors we've ever had.''
>
> B: (S3) 26-year-old Matt Smith has been cast as the next incarnation of the Doctor. Users on the Facebook Doctor Who forum that I frequent mostly had the same reaction: `` Who 's Matt Smith?''  (S4) **The guy is relatively unknown and the skeptics wondered if the right person was chosen.** (S5) After all, everyone speculated that Paterson Joseph, who had appeared on a couple episodes of the show, was going to be the next Doctor and here we get some no-name.
>
> We utilize the ES-comp approach to generate a discourse tree. Initially, we compress the three non-bolded sections S2, S3, and S5, resulting in the compressed texts C1, C2, and C3, respectively, as follows:
>
> C1: It's a sublime bit of casting.
>
> C2: Matt Smith has been cast as the next incarnation of the Doctor. ''Who 's Matt Smith?''
>
> C3: we get Paterson Joseph.
>
> Then, the compressed documents A and B are concatenated as [S1, C1, C2, S4, C3] and then are fed to a DRS Constructor. The resulting EDU sequence is displayed as follows:
>
> EDU1: Smith, 26,
>
> EDU2: who played a young political researcher in the show,
>
> EDU3: will become the biggest star of all
>
> EDU4: after winning the role of the 11th Doctor
>
> EDU5: It's a sublime bit of casting.
>
> EDU6: "Matt Smith has been cast as the next incarnation of the Doctor.
>
> EDU7: '' Who's Matt Smith ? ''
>
> EDU8: The guy is relatively unknown
>
> EDU9: and the skeptics wondered
>
> EDU10: if the right person was chosen
>
> EDU11: we get Paterson Joseph .
>
> Figure 4 (Page 8) illustrates the simplified discourse tree produced by the ES-comp strategy. Notably, we solely examine the coreferent relationship between "winning" and "chosen", hence Figure 4 excludes irrelevant nodes like EDU11. Furthermore, EDU1, EDU2, and EDU3 are merged into EDU1-3 since these three nodes are not part of DT-SDP. The DT-SDP in this case is [NS-Elaborate, NS-Elaborate, NS-Cause, SN-Span, SN-Cause].
>
> Directly measuring the similarity between EDU4 and EDU10 to predict coreference would result in misidentification due to their differing contexts. Since the rhetorical relation nodes "NS-Elaborate" in DT-SDP associate EDU4 with EDU1-3, EDU1-3 can introduce the agent "Smith" of the event mention "winning" to EDU4. "Smith" can also be passed to the event mention "chosen" in EDU10 using DT-SDP. This makes "Smith" apparent in the tokens of "winning" and "chosen" and facilitates accurate prediction.
>
> **Q3: I wonder what information this compressed document includes, and why it's better than full documents. The author (s) attribute this to noise in line 515, but some additional analysis here can be useful.**
>
> **A3**: The compressed file will preserve participant details of a particular event, aiding the enhancement of argument information. Examples can be seen in A2 above (namely C1, C2, and C3). This information is particularly valuable when making coreference predictions on two event mentions that share participants but differ in context as it boosts their similarity. The full documents contain extraneous event triggers and arguments. If constructing a discourse tree using the entire document as input, the model cannot distinguish the rhetorical relations between two specific event mentions. This can introduce additional errors that undermine the model's performance.
>
> Considering the examples in A2, the compressed text C1 omits information about other entities such as "The Guardian" and "Buchan", as well as redundant action words like "said" and "going". Removing these details allows the model to better focus on the rhetorical connection between the two specific event mentions "winning" and "chosen", which is helpful in predicting their relation.
>
> **Q4: In the ECB+ dataset, only few sentences from each document were annotated with coreference relations. To construct the discourse structure, did the author(s) use the full documents or only these annotated sentences?**
>
> **A4**: To create the discourse structure of event mention pairs within a document, we utilize the entire document as input. For event mention pairs across documents, we implement the ES-comp technique and also try feeding either the complete document or only annotated sentences to the DRS constructor. The experiment results are showed in table 4, which shows that ES-comp is the best strategy.
>
> **Q5: Following up on the above point, how would the discourse structure based method deal with longer documents. The related work CDLM (cited above) works on longer context inputs, so how do they compare against each other in long document settings.**
>
> **A5**: To deal with longer documents, we use a compression method to reduce their size while maintaining crucial data and removing irrelevant information. The experimental results on long documents show that our model using RoBERTa can achieve similar performance, in comparison with Longformer-based CDLM. This result indicates the effectiveness of the compression mechanism in filtering redundant information. If the RoBERTa encoder in our model is replaced by Longformer, our model will outperform CDLM, showed in the table of Q1.

---

### Meta-Review · Area_Chair_5ryj · 2023-09-19

**Recommendation:** 5

**Metareview:**

The paper presents a novel approach to incorporate global discourse structures for event coreference resolution, in particular, for cross-document event coreference resolution, the approach will conduct text compression on two documents except for two focus sentences that contain event mentions under scrutiny. Reviewers expressed excitements on this approach, especially on applying text compression to enable effective discourse analysis on the long meta document combining two original documents.

---

### Decision · Program_Chairs · 2023-10-07

**Decision:**

Accept-Main

**Comment:**

The paper presents a novel approach to incorporate global discourse structures for event coreference resolution, in particular, for cross-document event coreference resolution, the approach will conduct text compression on two documents except for two focus sentences that contain event mentions under scrutiny. Reviewers expressed excitements on this approach, especially on applying text compression to enable effective discourse analysis on the long meta document combining two original documents.